# ENCODING ONTOLOGIES WITH HOLOGRAPHIC RE-
# DUCED REPRESENTATIONS FOR TRANSFORMERS

## ABSTRACT

Transformer models trained on NLP tasks with medical codes often have randomly initialized embeddings that are then adjusted based on training data. For terms appearing infrequently in the dataset, there is little opportunity to improve these representations and learn semantic similarity with other concepts. Medical ontologies represent many biomedical concepts and define a relationship structure between these concepts, making ontologies a valuable source of domain-specific information. Holographic Reduced Representations (HRR) are capable of encoding ontological structure by composing atomic vectors to create structured higher-level concept vectors. Deep learning models can further process these structured vectors without needing to learn the ontology from training data. We developed an embedding layer that generates concept vectors for clinical diagnostic codes by applying HRR operations that compose atomic vectors based on the SNOMED CT ontology. This approach allows for learning the atomic vectors while maintaining structure in the concept vectors. We trained a Bidirectional Encoder Representations from the Transformers (BERT) model to process sequences of clinical diagnostic codes and used the resulting HRR concept vectors as the embedding matrix for the model. The HRR-based approach modestly improved performance on the masked language modeling (MLM) pre-training task (particularly for rare codes) as well as the fine-tuning tasks of mortality and disease prediction (particularly for patients with many rate codes). This method also supports explainability by separating representations of code-frequency, ontological information, and description words. This is the first time HRRs have been used to produce structured embeddings for transformer models and we find that this approach maintains semantic similarity between medically related concept vectors and allows better representations to be learned for rare codes in the dataset.

## 1 INTRODUCTION

Transformers (Vaswani et al., 2017) jointly optimize high-dimensional vector embeddings that represent input tokens, and a network that contextualizes and transforms these embeddings to perform a task. Originally designed for natural language processing (NLP) tasks, transformers are now widely used with other data modalities. In medical applications, one important modality consists of medical codes that are extensively used in electronic health records (EHR). A prominent example in this space is Med-BERT (Rasmy et al., 2021), which consumes a sequence of diagnosis codes. Tasks that Med-BERT and other EHR-transformers perform include disease and mortality prediction.

Deep networks have traditionally been alternatives to symbolic artificial intelligence with different advantages (Ganesan et al., 2021). Deep networks use real-world data effectively, but symbolic approaches have completive properties, such as better transparency and capacity for incorporating structured information. This has inspired many efforts to combine the two approaches in neuro-symbolic systems (Sarker et al., 2021).

Here we use a neuro-symbolic approach to incorporate structured knowledge from an authoritative medical ontology into transformer embeddings. Specifically, we encode concept relationships with a vector symbolic architecture to produce composite medical-code embeddings and backpropagate through the architecture to optimize the embeddings of atomic concepts. This approach produces optimized medical-code embeddings with an explicit structure that incorporates medical knowledge.

We test our method, HRRBERT, on the MIMIC-IV dataset (Johnson et al., 2022) and show improvements in both pre-training and fine-tuning tasks. We also show that our embeddings of ontologically similar rare medical codes have high cosine similarity, in contrast with embeddings that are learned in the standard way. Finally, we investigated learned representations of medical-code frequency, in light of recent demonstration of frequency bias in EHR-transformers (Yu et al., 2023).

We contribute:

- A novel neuro-symbolic architecture, HRRBERT, that combines vector-symbolic embeddings with the BERT architecture, leading to better performance in medical tasks.
- Efficient construction of vector-symbolic embeddings that leverage PyTorch autograd on GPUs.
- Optimized medical-code embeddings that better respect semantic similarity of medical terminology than standard embeddings for infrequently used codes.

## 1.1 BACKGROUND AND RELATED WORKS

Vector Symbolic Architectures (VSA), also called hyperdimensional computing, refer to a computing paradigm that relies on high dimensionality and randomness to represent concepts as unique vectors and perform operations in a high dimensional space (Kanerva, 2009). VSAs are cognitive models that create and manipulate distributed representations of concepts through the combination of base vectors with bundling, binding, and permutation algebraic operators (Gayler, 2004). For example, a scene with a red box and a green ball could be described with the vector SCENE=RED$\otimes$BOX+GREEN$\otimes$BALL, where $\otimes$ indicates binding, and $+$ indicates bundling. The atomic concepts of RED, GREEN, BOX, and BALL are represented by base vectors, which are typically random. VSAs also define an inverse operation that allows the decomposition of a composite representation. For example, the scene representation could be queried as SCENE$\otimes$BOX$^{-}1$. This should return the representation of GREEN or an approximation of GREEN that is identifiable when compared to a dictionary. In a VSA, the similarity between concepts can be assessed by measuring the distance between the two corresponding vectors.

VSAs were proposed to address challenges in modelling cognition, particularly language (Gayler, 2004). However, VSAs have been successfully applied across a variety of domains and modalities outside of the area of language as well, including in vision (Neubert & Schubert, 2021; Neubert et al., 2021), biosignal processing (Rahimi et al., 2019), and time-series classification (Schlegel et al., 2022). Regardless of the modality or application, VSAs provide value by enriching vectors with additional information, such as spatial semantic information in images and global time encoding in time series.

An early VSA framework proposed was Tensor Product Representation (Smolensky, 1990), which addressed the need for compositionality, but suffered from exploding model dimensionality. The VSA framework introduced by Plate (1995), Holographic Reduced Representations (HRR), improved upon Smolensky's by using circular convolution as the binding operator. Circular convolution keeps the output in the same dimension, solving the problem of exploding dimensionality.

In the field of deep learning, HRRs have been used in previous work to recast self-attention for transformer models (Alam et al., 2023), to improve the efficiency of neural networks performing a multi-label classification task by using an HRR-based output layer (Ganesan et al., 2021), and as a learning model itself with a dynamic encoder that is updated through training (Kim et al., 2023). In all of these works, the efficiency and simple arithmetic of HRRs are leveraged. Our work differs in that we also leverage the ability of HRRs to create structured vectors to represent complex concepts as inputs to a transformer model.

VSAs such as HRRs can effectively encode domain knowledge, including complex concepts and the relationships between them. For instance, Nickel et al. (2015) propose holographic embeddings that make use of VSA properties to learn and represent knowledge graphs. Encoding domain knowledge is of interest in the field of deep learning, as it could improve, for example, a deep neural network's ability to leverage human knowledge and to communicate its results within a framework that humans understand (Dash et al., 2021). Ontologies are a form of domain knowledge that have been incorporated into machine learning models to use background knowledge to create embeddings with meaningful similarity metrics and for other purposes (Kulmanov et al., 2020). In our work, we

use HRRs to encode domain knowledge in trainable embeddings for a transformer model. The domain knowledge we use comes from the Systematized Medical Nomenclature for Medicine–Clinical Terminology (SNOMED CT)—a popular clinical ontology system that includes definitions of relationships between clinical concepts (Riikka et al., 2023). Additional information on SNOMED CT is provided in Appendix A.4.

To the best of our knowledge, HRRs have not been used before as embeddings for transformer models. Transformer models typically use learned embeddings with random initializations (Vaswani et al., 2017). However, in the context of representing ontological concepts, using such unstructured embeddings can have undesirable effects. One problem is the inconsistency between the rate of co-occurrence or patterns of occurrence of medical concepts and their degree of semantic similarity described by the ontology. For example, the concepts of "Type I Diabetes" and "Type II Diabetes" are mutually exclusive in EHR data and do not follow the same patterns of occurrence due to differences in pathology and patient populations (Song et al., 2019). The differences in occurrence make it difficult for a transformer model to learn embeddings with accurate similarity metrics. The concepts should have relatively high similarity according to the ontology. They both share a common ancestor of "Diabetes Mellitus," they are both metabolic disorders that affect blood glucose levels, and they can both lead to similar health outcomes. Song et al. (2019) seeks to address this type of inconsistency by training multiple "multi-sense" embeddings for each non-leaf node in an ontology's knowledge graph via an attention mechanism. However, the "multi-sense" embeddings do not address the learned frequency-related bias that also arises from the co-occurrence of concepts. Frequency-related bias raises an explainability issue, as it leads to learned embeddings that do not reflect true similarity relationships between concepts, for example, as defined in an ontology, but instead reflect the frequency of the concepts in the dataset (Yu et al., 2023). This bias particularly affects codes that are used less frequently.

Our proposed approach, HRRBERT, uses SNOMED CT to represent thousands of concepts with high-dimensional vectors such that each vector reflects a particular clinical meaning and can be compared to other vectors using the HRR similarity metric, cosine similarity. It also leverages the computing properties of HRRs to provide a fault-tolerant and robust framework for the structured embeddings of the transformer model that supports optimization through backpropagation.

## 2 METHODS

### 2.1 MIMIC-IV DATASET

The data used in this study was derived from the Medical Information Mart for Intensive Care (MIMIC) v2.0 database, which is composed of de-identified EHRs from in-patient hospital visits between 2008 and 2019 (Johnson et al., 2022). MIMIC-IV is available through PhysioNet (Goldberger et al., 2000). We used the ICD-9 and ICD-10 diagnostic codes from the *icd_diagnosis* table from the MIMIC-IV *hosp* module. We filtered patients who did not have at least one diagnostic code associated with their records. Sequences of codes were generated per patient by sorting their hospital visits by time. Within one visit, the order of codes from the MIMIC-IV database was used, since it represents the relative importance of the code for that visit. Each unique code was assigned a token. In total, there were 189,980 patient records in the dataset. We used 174,890 patient records for pre-training, on which we performed a 90-10 training-validation split. We reserved 15k records for fine-tuning tasks.

### 2.2 ENCODING SNOMED ONTOLOGY WITH HRR EMBEDDINGS

In this section, we detail the methodologies of constructing vector embeddings for ICD disease codes using HRR operations based on the SNOMED CT structured clinical vocabulary. We first describe our mapping from ICD concepts to SNOMED CT terms. Next, we define how the atomic symbols present in the SNOMED CT ontology are combined using HRR operations to construct concept vectors for the ICD codes. Finally, we describe our method to efficiently compute the HRR embedding matrix using default PyTorch operations that are compatible with autograd.

### 2.2.1 MAPPING ICD TO SNOMED CT ONTOLOGY

Our data uses ICD-9 and ICD-10 disease codes while our symbolic ontology is defined in SNOMED CT, so we required a mapping from the ICD to the SNOMED CT system to build our symbolic architecture. We used the SNOMED CT International Release from May 31, 2022 and only included SNOMED CT terms that were active at the time of that release. While SNOMED publishes a mapping tool from SNOMED CT to ICD-10, a majority of ICD-10 concepts have one-to-many mappings in the ICD-to-SNOMED CT direction (NLM, 2022b). To increase the fraction of one-to-one mappings, we used additional published mappings from the Observational Medical Outcomes Partnership (OMOP) (OHDSI, 2019), mappings from ICD-9 directly to SNOMED CT (NLM, 2022a), and mappings from ICD-10 to ICD-9 (NCHS, 2018). Specific details on how these mappings were used can be found in Appendix A.8.

Notably, after excluding ICD codes with no active SNOMED CT mapping, 671 out of the 26,164 unique ICD-9/10 codes in the MIMIC-IV dataset were missing mappings. When those individual codes were removed, a data volume of 4.62% of codes was lost. This removed 58 / 190,180 patients from the dataset, as they had no valid ICD codes in their history. Overall, the remaining 25,493 ICD codes mapped to a total of 12,263 SNOMED CT terms.

### 2.2.2 SNOMED CT VECTOR SYMBOLIC ARCHITECTURE

Next, we define how the contents of the SNOMED CT ontology were used to construct a symbolic graph to represent ICD concepts. For a given SNOMED CT term, we used its descriptive words and its relationships to other SNOMED CT terms. A relationship is defined by a relationship type and a target term. In total, there were 13,852 SNOMED CT target terms and 40 SNOMED CT relationship types used to represent all desired ICD concepts. In the ontology, many ICD concepts share SNOMED CT terms in their representations.

The set of relationships was not necessarily unique for each SNOMED CT term. To add more unique information, we used a term's "fully specified name" and any "synonyms" as an additional set of words describing that term. We set all text to lowercase, stripped punctuation, and split on spaces to create a vocabulary of words. We removed common English stopwords from a custom stopword list that was collected with assistance from a medical physician. The procedure resulted in a total of 8833 vocabulary words.

Overall, there were a total of 22,725 "atomic" symbols for the VSA which included the SNOMED CT terms, relationships, and the description vocabulary. Each symbol was assigned an "atomic vector". We built a "concept vector" for each of the target 25,493 ICD codes using HRR operations to combine atomic vectors according to the SNOMED CT ontology structure.

To build a $d$-dimensional concept vector for a given ICD concept, we first considered the set of all relationships that the concept maps to. We used the HRR operator for binding, circular convolution, to combine vectors representing the relationship type and destination term together and defined the concept vector to be the bundling of these bound relationships. For the description words, we bundled the vectors representing each word together and bound this result with a new vector representing the relationship type "description".

$$\boldsymbol{x}_{\text{ICD concept}} = \sum_{\text{SNOMED CT}} \boldsymbol{x}_{\text{relationship}} \circledast \boldsymbol{x}_{\text{term}} + \sum_{\text{words}} \boldsymbol{x}_{\text{description}} \circledast \boldsymbol{x}_{\text{word}}$$

Formally, let $\mathbb{A} : \{1, 2, ..., N_a\}$ be the set of integers enumerating the unique atomic symbols for SNOMED CT terms and description words. Let $\mathbb{B} : \{1, 2, ..., N_r\}$ be the set of integers enumerating unique relationships for SNOMED CT terms, including the description relationship and the binding identity. Let $\mathbb{D} : \{1, 2, ..., N_c\}$ be the set of integers enumerating the ICD-9 and ICD-10 disease concepts represented by the VSA.

$\mathbb{A}$ has an associated embedding matrix $\boldsymbol{A} \in \mathbb{R}^{N_a \times d}$, where atomic vector $\boldsymbol{a}_k = \boldsymbol{A}_{k,:}, k \in \mathbb{A}$ is the $k$-th row the embedding matrix. Similarly, there is relationship embedding matrix, $\boldsymbol{R} \in \mathbb{R}^{N_r \times d}$ and $\boldsymbol{r}_j = \boldsymbol{R}_{j,:}, j \in \mathbb{B}$; and an ICD concept embedding matrix, $\boldsymbol{C} \in \mathbb{R}^{N_c \times d}$ and $\boldsymbol{c}_i = \boldsymbol{C}_{i,:}, i \in \mathbb{D}$. We described the VSA with the following formula, where $\mathcal{G}_i$ is a graph representing the connections between ICD concept $i$ to atomic symbols $k$ by relationship $j$.

$$c_i = \sum_{(j,k) \in \mathcal{G}_i} r_j \circledast a_k$$

## 2.3 LEARNING THROUGH HRR OPERATIONS

To make the HRR concept embeddings useful for a deep neural network, the operations used to form the embeddings need to be compatible with backpropagation so that gradient descent can update the lower-level atomic vectors. We desired a function that produced the ICD concept embedding matrix, $C$, given the inputs of the VSA knowledge graphs, $\mathcal{G}_i$, and symbol embedding matrices, $R$ and $A$.

We described our earlier attempts in Appendix A.6, but the main outcome was that our initial implementation was impractical, as backpropagation either ran out of memory or required too long to compute to be practical, i.e. over 1 minute. This motivated the need for a more efficient algorithm to compute our HRR embedding matrix. Our final approach took advantage of the fact that many disease concepts use a relationship type, but to different atomic symbols. Thus, for a particular relationship type, we could contribute to building many disease concept vectors at once by selecting many atomic vectors, doing a vectorized convolution with the relationship vector, and distributing the results to be added with the appropriate concept embedding rows. This step needed to be repeated at most $m$ times for a particular relationship, where $m$ is the maximum multiplicity of that relationship among all concepts. This approach was still fast on GPUs since we could vectorize matrix indexing and circular convolution operations.

We improved memory efficiency by performing fast Fourier transforms (FFTs) on the atomic vector embeddings and construct the concept vectors by performing binding via element-wise multiplication in the Fourier domain. Due to the linearity of the HRR operations, we performed a final FFT on the complex-valued concept embedding to convert back to the real domain. This saved both time and memory since each circular convolution would have needed to perform two FFT operations, but we combined those all at the beginning and end of the algorithm.

The final algorithm used ~3.5 GB of memory and took ~80 ms and ~550 ms for forward and backward pass respectively on a single RTX 3090 GPU for the HRR embedding matrix with MIMIC-IV codes. This method was written with standard PyTorch operations using tensor indexing, addition, element-wise multiplication, and FFT operations, allowing us to use PyTorch autograd to backpropagate through the HRR operations and update atomic vectors with gradient descent.

### 2.3.1 EMBEDDING CONFIGURATIONS

We call our method of constructing embeddings for ICD codes purely from HRR representations "HRRBase" and the standard method of creating transformer token embeddings from random vectors "unstructured". While the HRRBase configuration enforces the ontology structure, we wondered whether it would be too rigid and have difficulty representing information not present in SNOMED CT. As dataset frequency information for ICD medical codes is not present in the HRR structure, we tried some alternate configurations by adding an embedding that represented the empirical frequency of that ICD code in the dataset. We also tried adding fully learnable embeddings with no prior structure.

Given the wide range of ICD code frequencies in MIMIC, we log-transformed the empirical ICD code frequencies, and then discretized the resulting range. For our HRRFreq configuration, we used the sinusoidal frequency encoding as in Vaswani et al. (2017) to encode the discretized log-frequency information. The frequency embeddings were normalized before being summed with the HRR embedding vectors.

We defined two additional configurations in which a standard embedding vector was integrated with the structured HRR concept vector. With "HRRAdd", a learnable embedding was added to the concept embedding, HRRAdd = $C + L_{\text{add}}$, $L_{\text{add}} \in \mathbb{R}^{N_c \times d}$. However, this roughly doubled the number of learnable parameters compared to other formulations.

With "HRRCat", a learnable embedding of dimension $d/2$ was concatenated with the HRR concept embedding of dimension $d/2$. This keeps the total number of learnable parameters was roughly the same as the unstructured configuration (25,493 $d$-dimensional vectors) and the HRRBase con-

figuration (22,725 $d$-dimensional vectors). The final embedding matrix was defined as HRRCat = $[C\ L_{\text{cat}}]$, where $C, L_{\text{cat}} \in R^{N_c \times d/2}$.

## 2.4 MODEL ARCHITECTURE

We utilized a BERT-base model architecture with a post-layer norm position and a sequence length of 128 ICD codes (Devlin et al., 2018). A custom embedding class was used to support the functionality required for our HRR embeddings. We adapted the BERT segment embeddings to represent groups of codes from the same hospital visit, using up to 100 segment embeddings to encode visit sequencing. An embedding dimension of $d = 768$ was used, and all embeddings were initialized from $x \sim \mathcal{N}_d(0, 0.02)$, as in Devlin et al. (2018), including the atomic vectors for HRR embeddings. Fine-tuning used a constant learning rate schedule with a weight decay of 4e-6. Fine-tuning lasted 10 epochs with a batch size of 80.

## 2.5 EXPERIMENTS

We pre-trained HRRBERT with 3 trials each for the unstructured, HRRBase, HRRCat, and HRRAdd embedding configurations. For each of the 3 pre-trained models, 10 fine-tuning trials were conducted for a total of 30 trials per fine-tuning task. The best checkpoint from the 10 epochs of fine-tuning was saved based on validation performance. A test set containing 666 patient records was used to evaluate each of the fine-tuned models for both mortality and disease prediction. We report accuracy, precision, recall, and F1 scores averaged over the 30 trials for the fine-tuning tasks.

# 3 EXPERIMENTAL RESULTS

## 3.1 PRE-TRAINING

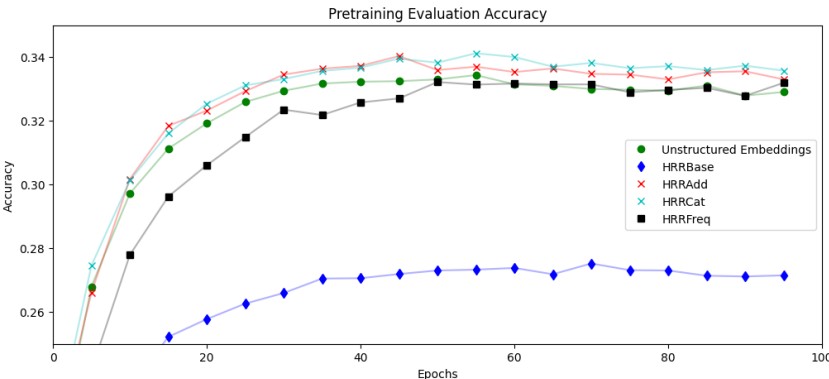

Figure 1: Pre-training validation set evaluation results for different configurations

MLM accuracy is evaluated on a validation set over the course of pre-training. Pre-training results for different configurations are shown in Figure 1. The pre-training results are averaged over 3 runs for each of the configurations except for HRRFreq where only 1 model run was completed.

The baseline of learned unstructured embeddings has a peak pre-training validation performance of around 33.4%. HRRBase embeddings perform around 17% worse compared to the baseline of learned unstructured embeddings. We hypothesize that this decrease in performance is due to a lack of embedded frequency information in HRRBase compared to learned unstructured embeddings. HRRFreq (which combines SNOMED CT information with frequency information) has a similar performance compared to unstructured embeddings, supporting this hypothesis. Compared to baseline, HRRAdd and HRRCat improve pre-training performance by a modest margin of around 2%. We posit that this almost 20% increase in performance of HRRCat and HRRAdd over HRRBase during pre-training is partly due to the fully learnable embedding used in HRRCat and HRRAdd learning frequency information.

Table 1: Finetuning mean test scores and standard deviations for both mortality and disease prediction tasks. The best scores are bolded.

| Finetuning Task | Configuration | Accuracy | Precision | Recall | F1-Score |
|---|---|---|---|---|---|
| Mortality Prediction | HRRBase | **84.4**±2.3 | **65.8**±2.0 | 85.6±2.2 | **69.2**±2.7 |
| | HRRAdd | 84.0±2.2 | 65.7±1.9 | **85.7**±2.3 | 68.9±2.5 |
| | HRRCat | 83.9±2.3 | 65.6±1.7 | 84.9±2.8 | 68.8±2.5 |
| | Unstructured | 83.4±1.9 | 64.9±1.2 | 84.6±2.2 | 67.9±1.8 |
| Disease Prediction | HRRBase | **79.9**±0.5 | **73.0**±1.2 | 67.2±0.7 | 69.0±0.6 |
| | HRRAdd | 79.6±0.7 | 72.6±1.4 | 67.3±0.9 | 69.0±0.6 |
| | HRRCat | 79.6±0.8 | 72.5±1.7 | 67.3±1.0 | 68.9±0.8 |
| | Unstructured | 79.4±0.5 | 72.1±1.1 | **67.8**±1.0 | **69.2**±0.7 |

## 3.2 FINE-TUNING

**Mortality Prediction Task:** The mortality prediction task is defined as predicting patient mortality within a 6-month period after the last visit. Binary mortality labels were generated by comparing the time difference between the last visit and the mortality date. A training set of 13k patient records along with a validation set of 2k patient records were used to fine-tune each model on mortality prediction. Table 1 shows the evaluation results of mortality prediction for each of the configurations. We performed a two-sided Dunnett's test to compare our multiple experimental HRR embedding configurations to the control unstructured embeddings, with $p < 0.05$ significance level. Levene's test shows that the equal variance condition is satisfied, and the Shapiro-Wilk test suggests normal distributions except for HRRBase precision and F1. No comparisons of mean metrics for HRR embeddings were significantly greater than the control. An additional experiment conducted on the Philips Electronic Intensive Care Unit (eICU) (Pollard et al., 2018) shows corroborating results with the MIMIC-IV experiments. The details of the eICU experiments are reported in Appendix A.1. We applied our mortality prediction model fine-tuned on MIMIC-IV to the eICU dataset without further training. We found that HRRBase embeddings had a significantly greater mean accuracy ($p = 0.046$) compared to unstructured embeddings when applied to the eICU dataset.

**Disease Prediction Task:** The disease prediction task is defined as predicting which disease chapters were recorded in the patient's last visit using information from earlier visits. We converted all ICD codes in a patient's last visit into a multi-label binary vector of disease chapters. As there are 22 disease chapters defined in ICD-10, the multi-label binary vector has a size of 22 with binary values corresponding to the presence of a disease in each chapter. A training set of 4.5k patient records along with a validation set of 500 patient records were used to fine-tune each model on this task. Table 1 shows the evaluation results of disease prediction for each of the configurations. For the two-sided Dunnett test, Levene's test shows that the equal variance condition is satisfied, and the Shapiro-Wilk test suggests normal distributions except for HRRAdd accuracy. The test showed HRRBase embeddings had a significantly greater mean accuracy ($p = 0.033$) and precision ($p = 0.023$) compared to unstructured embeddings. No other comparisons of mean metrics for HRR embeddings were significantly greater than the control. An additional experiment conducted with the disease prediction task is detailed in Appendix A.3. We constructed a really-out-of-distribution (ROOD) dataset from MIMIC-IV patients whose contexts contain at least one of 32 codes that were never used to train HRRBase and unstructured models. We found that HRRBase strongly outperforms unstructured for the disease prediction task on patients with only unseen codes (F1 Score 79.5 vs. 48.0). We suspect that the shared embedding components in HRRBase contribute this increase.

## 3.3 T-SNE AND COSINE SIMILARITY

We computed t-SNE dimension reductions to visualize relationships among ICD code embeddings in the pre-trained models. Figure 2 shows that unstructured embeddings of common ICD codes are clustered together with a large separation from those of uncommon codes. This suggests that code-frequency information is prominently represented in these embeddings, consistent with frequency bias in related models (Yu et al., 2023). Common and uncommon code clusters are less distinct

in HRRBase, which does not explicitly encode frequency information. A t-SNE analysis of the unstructured components of the HRRCat and HRRAdd embeddings, provided in Appendix A.5, suggests that these additional unstructured embeddings learn some frequency information.

We further tested frequency bias by masking individual codes of different frequencies in the MLM validation dataset and measured the top-10 MLM inference accuracy, further described in Appendix A.2. The results from this experiment show that HRRAdd and HRRCat embedding models have a significantly greater rare-code prediction accuracy compared to unstructured embedding models.

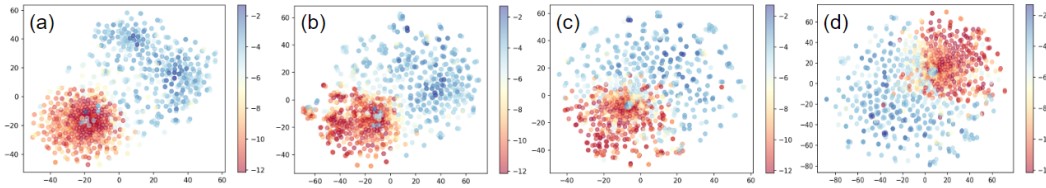

Figure 2: Comparing t-SNE of (a) unstructured embeddings, (b) HRRAdd, (c) HRRCat, and (d) HRRBase. The t-SNE graphs are color-coded by the frequency of the ICD codes in the dataset - highly frequent codes are colored blue while infrequent codes are colored red.

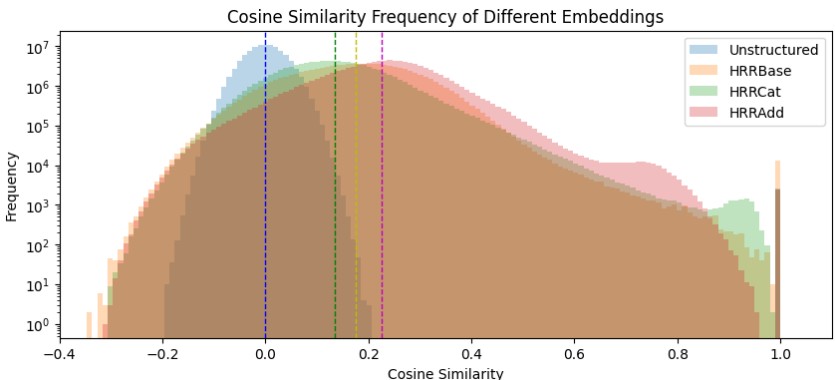

Figure 3: The sample distribution of cosine similarities for different embeddings. Each sample distribution is based on a sample of 100 million cosine similarity points from over 1 billion total cosine similarity points for each embedding configuration. The average cosine similarity values are marked by dotted lines for each configuration.

We also computed the distributions of cosine similarities of the embeddings for each of the configurations and conducted a small case study to understand the cosine similarities of a few codes and how they reflect medical relationships. Figure 3 shows the distribution of cosine similarities for each configuration's embeddings. The HRR configurations produced a much broader distribution of cosine similarities compared to unstructured embeddings, including much higher similarities.

Table 2 shows a case study for codes *Other and unspecified hyperlipidemia* (2724-9), *Hypothermia* (9916-9), and *Gastro-esophageal Reflux disease without esophagitis* (K219-10). In the first case study for 2724-9, we observe highly ontologically similar codes, such as *Other hyperlipidemia* and *Hyperlipidemia, unspecified*, are encoded with high cosine similarity for HRRBase, which is not the case for unstructured embeddings. The co-occurrence problem can be seen in the second case study for 9916-9. The most similar codes for HRRBase are medically similar codes that would not usually co-occur, while for unstructured embeddings the most similar codes co-occur frequently. For the final case study on K219-10, frequency-related bias can be observed in the unstructured embeddings with frequent but mostly ontologically unrelated codes as part of the top list of cosine similar codes, whereas the top list of cosine similar codes for HRRBase contains medically similar codes.

We broadened this case study to test statistical differences in cosine and semantic embedding similarity between structured and unstructured embeddings. 30 ICD codes were selected from different

Table 2: Three cosine similarity case studies looking at related ICD codes for unstructured and HRRBase. The top 4 cosine-similar ICD codes to the chosen code are listed (most to least similar) with their full description and similarity value.

| 2724-9 - Other and unspecified hyperlipidemia | | | |
|---|---|---|---|
| Unstructured | | HRRBase | |
| Pure hypercholesterolemia | 0.542 | Other hyperlipidemia | 1.000 |
| Hyperlipidemia, unspecified | 0.482 | Hyperlipidemia, unspecified | 1.000 |
| Esophageal reflux | 0.304 | Pure hypercholesterolemia | 0.463 |
| Anemia, unspecified | 0.279 | Mixed hyperlipidemia | 0.418 |
| 9916-9 - Hypothermia | | | |
| Unstructured | | HRRBase | |
| Frostbite of hand | 0.418 | Hypothermia, initial encounter | 0.794 |
| Frostbite of foot | 0.361 | Hypothermia not with low env. temp. | 0.592 |
| Drowning and nonfatal submersion | 0.352 | Effect of reduced temp., initial encounter | 0.590 |
| Immersion foot | 0.341 | Other specified effects of reduced temp. | 0.590 |
| K219-10 - Gastro-esophageal reflux disease without esophagitis | | | |
| Unstructured | | HRRBase | |
| Esophageal reflux | 0.565 | Esophageal reflux | 0.635 |
| Hyperlipidemia, unspecified | 0.335 | Gastro-eso. reflux d. with esophagitis | 0.512 |
| Anxiety disorder, unspecified | 0.332 | Reflux esophagitis | 0.512 |
| Essential (primary) hypertension | 0.326 | Hypothyroidism, unspecified | 0.268 |

frequency categories in the dataset, with 10 codes drawn randomly from the 300 most common codes, 10 codes drawn randomly by weighted frequency from codes appearing fewer than 30 times in the dataset, and 10 codes randomly selected by weighted frequency from the entire dataset. For each selected code, the top 4 cosine-similar ICD codes were assessed by a physician for ontological similarity. For each frequency category, a one-tailed Fisher's exact test was conducted with a significance level of $p < 0.05$ to determine whether a relationship existed between embedding type and clinical relatedness. Only the rare codes result were statistically significant, with $p = 2.44 \times 10^{-8}$. This suggests that structured embeddings are associated with greater clinical relevance of the top cosine-similar codes than unstructured embeddings.

## 4 CONCLUSION

We proposed a novel hybrid neural-symbolic approach called HRRBERT that integrates medical ontologies represented by HRR embeddings. In tests with the MIMIC-IV dataset, HRRBERT models modestly outperformed baseline models with unstructured embeddings for pre-training, disease prediction accuracy, and mortality prediction F1. HRRBERT models had pronounced performance advantages in MLM with rare codes and disease prediction for patients with no codes seen during training. We also showed that HRRs can be used to create medical code embeddings that better respect ontological similarities for rare codes. A key benefit of our approach is that it facilitates explainability by disentangling token-frequency information, which is prominently represented but implicit in unstructured embeddings. Critical to this approach is a new method to construct vector-symbolic embeddings that leverage PyTorch autograd on GPUs, allowing learning through HRR operations. Limitations to address in future work include the complexity of processing knowledge graphs to be compatible with HRRs. Our method also relies on rare code HRRs sharing atomic elements with common code HRRs. However, rare codes are also likely to contain rare atomic elements. This suggests a method improvement to incorporate more shared information into rare atomic elements. For example, we could initialize description word vectors with pre-trained language model embeddings. Because HRRs can be queried with linear operations, future work could also explore whether transformers can learn to extract specific information from these composite embeddings.

## 5 REPRODUCIBILITY

The supplementary materials include all code to prepare the data from raw MIMIC-IV files, pre-train the models, and fine-tune the models. Independent access to MIMIC-IV data is necessary as its license does not allow redistribution. Exact instructions for reproduction can be found in a README file in the supplementary materials.

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

## A  APPENDIX

### A.1  eICU OUT-OF-DISTRIBUTION RESULTS

The Philips Electronic Intensive Care Unit (eICU) (Pollard et al., 2018) is a collaborative dataset that covers patients who were admitted to critical care units in 2014 and 2015. The eICU dataset is comprised of 200k de-identified health data including vital signs measurements, care plan documentation, the severity of illness measures, diagnosis information, and treatment information. Patient data is collected from one of 334 units at 208 hospitals located in the US.

We used the ICD-9 and ICD-10 codes from the *diagnosis* table and the visit information from the *patient* table in the eICU dataset to build sequences of diagnostic codes within each hospital visit. Each visit contains several ICD-9 and ICD-10 codes, although many of the ICD-9 and ICD-10 codes represent the same diagnosis. Because the column was named *icd9code*, we prioritized taking only ICD-9 codes and used ICD-10 codes when no ICD-9 codes were available. All codes selected this way had a valid VSA representation from the MIMIC-IV model.

We separated visits based on unique *patienthealthsystemstayid* keys. We sorted the codes within a visit first by decreasing the priority class, and then from earliest to latest within the same priority class. Unlike in MIMIC-IV, there is no way to guarantee that different hospital visits could be ordered by time to create a sequence of visits, so we were limited to making predictions within the context of one visit. We could therefore only do our mortality prediction task, since our disease prediction task requires at least two visits in a patient's history.

For our experiment, we applied our mortality prediction models that were fine-tuned on MIMIC-IV to eICU data to see if our results generalize. These models are not optimized for mortality prediction for other hospitals where coding methodology and clinical practice may differ. For example, the most common code in the eICU dataset represents acute respiratory failure, whereas the most common code in the MIMIC-IV dataset represents hypertension.

Results from Table 3 corroborate results from Table 1 showing that HRRBase modestly outperforms unstructured embeddings for mortality prediction on the eICU dataset without further fine-tuning.

Table 3: Finetuning mean test scores and standard deviations for mortality prediction on the eICU dataset. The best scores are bolded.

| Finetuning Task | Configuration | Accuracy | Precision | Recall | F1-Score |
|---|---|---|---|---|---|
| eICU Mortality Prediction | HRRBase | **68.9**±1.3 | **75.0**±1.8 | **57.0**±5.8 | **64.5**±3.5 |
| | HRRAdd | 68.1±1.6 | 74.0±2.2 | 56.2±6.8 | 63.6±3.9 |
| | HRRCat | 68.2±1.2 | 73.8±2.6 | 57.0±7.2 | 64.0±3.7 |
| | Unstructured | 68.0±1.4 | 74.0±2.6 | 56.0±7.0 | 63.4±3.9 |

For a two-sided Dunnett's test, Levene's test shows that the equal variance condition is satisfied, and the Shapiro-Wilk test suggests normal distributions except for HRRAdd accuracy and HRRBase accuracy, recall, and precision. Applying the test, we find that HRRBase accuracy is statistically greater ($p = 0.046$) when compared to the unstructured embedding performance on the mortality prediction task on eICU.

## A.2 Top-k Accuracy for MLM

Given that the model trains and sees more common codes compared to rare codes, rare codes are naturally challenging to predict. Rare code prediction is a contradiction, where performance can only be increased by having more examples of the rare code (Wang et al., 2022). Different DL approaches have been designed to tackle the rare code problem in MIMIC by integrating code hierarchy, co-occurrences, joint learning architectures, and embedding medical concepts from clinical notes and Wikipedia documents (Wang et al., 2022; Vu et al., 2020; Cao et al., 2020).

We hypothesized that our HRR embedding models should have improved accuracy when predicting rare codes in the dataset compared to unstructured embedding models, since rare codes should share some atomic vectors in their representations with common codes. To test this, we evaluated the accuracy of an MLM pre-trained model predicting a single masked code of a known frequency. We split the codes in the pre-training validation dataset into 7 bins from log frequency -14 to 0, such that each bin has a width of 2. The most common codes are in a bin with log frequencies between -2 and 0, while the rarest codes are from a bin with log frequencies between -14 and -12. From each bin, we selected 400 codes at random, repeating codes from that bin if there were fewer than 400. For each of these codes, we select one patient that had that code in their history, mask that code as would be done in MLM, and create a dataset of these 2,800 patients to use for MLM inference. Given the top-1 accuracy of rare codes is extremely poor, we expanded to the top-k accuracy for $k = 2, 3, 5, 10, 100$. This is also motivated by the fact that some of the embedding representations in HRRBase may be identical to each other, given the mapping scheme. This may result in a classifier having a difficult time determining the best output class if there are many similar vectors in the final linear layer.

Figure 5 shows the MLM top-10 accuracy on predicting codes in the different frequency bins, averaged across the three pre-training models per configuration. Significant comparisons to the unstructured control at a $p < 0.05$ level indicated with an asterisk. We assess statistical significance for each bin using a two-tailed Dunnett's test comparing mean accuracy scores of experimental HRR configurations against the control unstructured configuration. Levene's test shows the measurements meet the equal variance assumption, and the Shapiro-Wilk test suggest that measurements are normally distributed, except for the following: for top-10 accuracy HRRAdd (bins -6 and -10), HRRCat (bins -4, -10, and -12), HRRBase (bins -8 and -12), and unstructured (bin -8); and for top-100 accuracy HRRAdd (bins 0 and -4) and HRRBase (bin -4).

In frequency bin -8, the mean top-10 accuracy of HRRAdd and HRRCat are significantly greater than the mean accuracy of unstructured, $p = 0.003, p = 0.015$ respectively. In frequency bin -10, the mean top-10 accuracy of HRRAdd and HRRBase are significantly greater than the mean accuracy of unstructured, $p = 0.004, p = 0.033$ respectively. This trend in significantly greater top-100 accuracy for HRR embeddings compared to unstructured embeddings on rare codes is further shown in Figure 6. Notably, the top-100 accuracy in frequency bin -12 is non-zero for the HRR methods. These codes in the rarest bin occur only once in the dataset and therefore have never been used by the model for gradient updates, since they are in the validation dataset. This suggests that

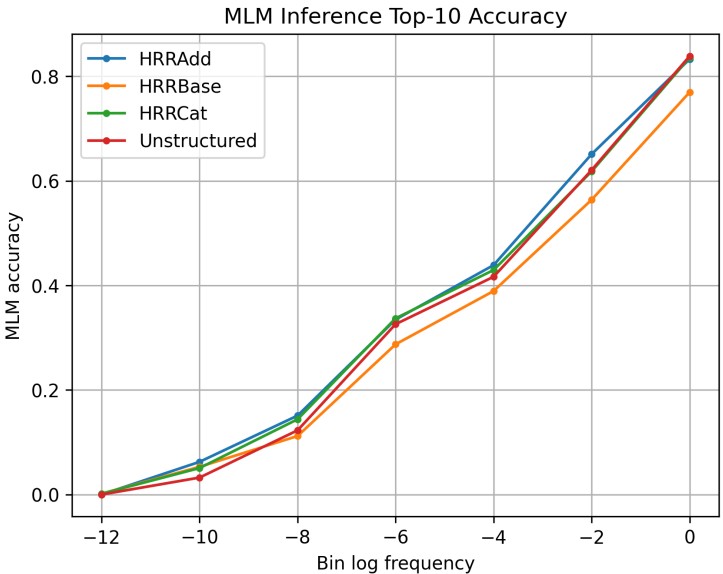

Figure 4: The top-10 MLM accuracy versus bin log frequency for different embedding configurations.

the HRR methods have some ability to provide clinically relevant information to rare codes masked disease prediction, though only when given the ability to consider more than one possible label.

Although HRRBase embeddings have significantly lower top-10 and top-100 accuracy scores compared to unstructured embeddings, we suspect is because frequency information is not explicitly encoded in HRRBase embeddings, which might be important for good performance on this masked code prediction task. This corroborates with the fact that HRRBase MLM accuracy in pre-training is a lot lower than the other configurations. However, when frequency information is explicitly added, as in HRRFreq, the performance of the HRR embeddings improves.

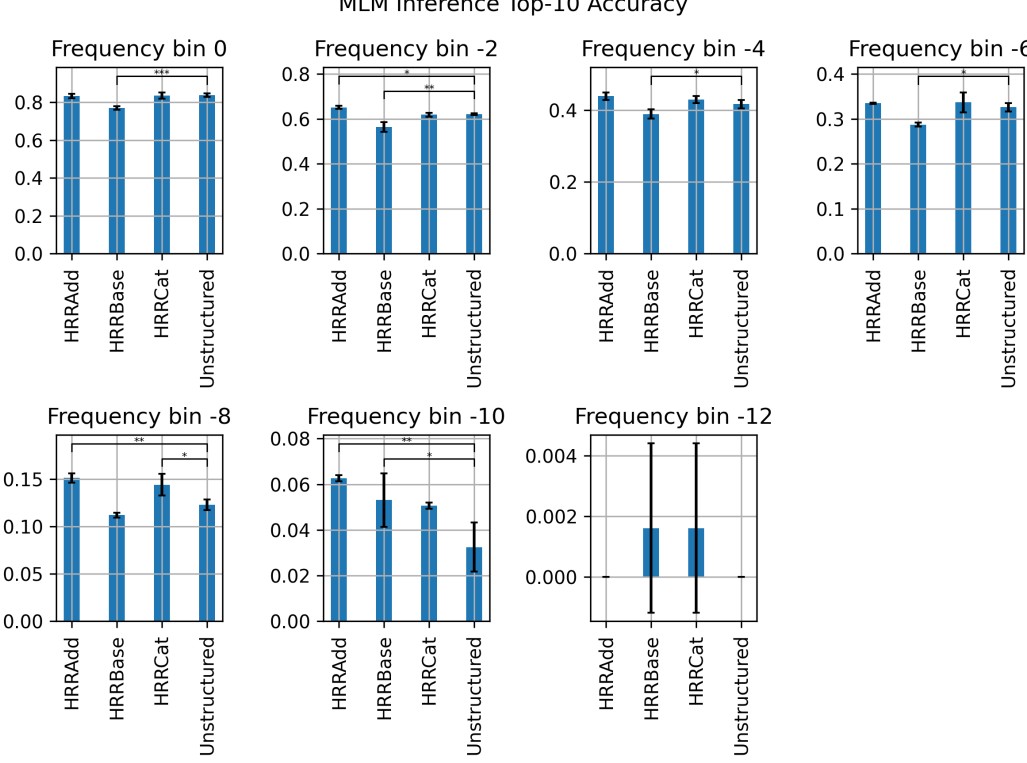

Figure 5: The top-10 MLM accuracy for binned code frequencies in log scale. Common codes are in frequency bin 0 with rarest codes being in frequency bin -12. 0.05, 0.01, and 0.001 significance levels comparing to unstructured embeddings are indicated with 1, 2, and 3 asterisks respectively..

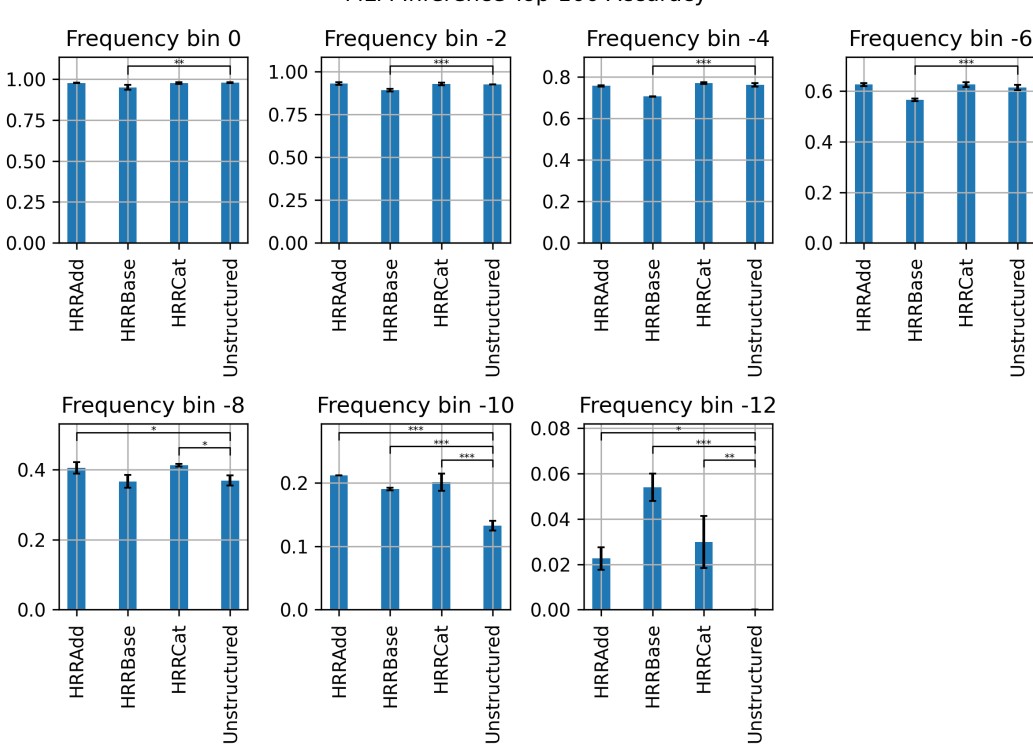

Figure 6: The top-100 MLM accuracy for binned code frequencies in log scale. Common codes are in frequency bin 0 with rarest codes being in frequency bin -12. 0.05, 0.01, and 0.001 significance levels comparing to unstructured embeddings are indicated with 1, 2, and 3 asterisks respectively.

### A.3 REALLY-OUT-OF-DISTRIBUTION RESULTS

For this experiment, we created and used a dataset denoted as really-out-of-distribution (ROOD). The ROOD dataset consists of nearly 30k patient records from MIMIC-IV that contain at least one of 32 ICD codes. These 32 codes were selected from 6 patients whose records fully consist of these codes. The 32 selected ICD codes are contained in Appendix A.7. Given that these codes appear in a substantial number of total records ( 16% of MIMIC-IV), we can see that it is not the case that all 32 codes are exceedingly common or rare. However, as these codes make up only 3.5% of all codes in the ROOD dataset and are not strongly associated with mortality, we only report results on the disease prediction task.

We create pre-training and fine-tuning datasets that contain records in MIMIC-IV that have no ROOD codes. HRRBase and unstructured embedding models are then pre-trained using MLM to learn embedding representations of codes and fine-tuned for disease prediction on these cleaned datasets. Thus, during training, the HRRBase and unstructured models do not encounter any examples using the 32 ROOD codes and so do not explicitly learn representations for those codes. The trained models are then tested using the ROOD dataset.

Results from Table 4 on ROOD dataset disease prediction show that HRRBase outperforms the unstructured embedding model for contexts of entirely unseen codes. We assess statistical significance using two-tailed, independent t-test with unequal variance, as some measurements failed Levene's test for equal variance. The means of all the metrics for HRRBase are significantly greater than for unstructured when making inferences on patients with entirely unseen codes, $p < 0.001$ for all metrics. Given the embedded ontological structure, we hypothesize that HRRBase implicitly learns useful embeddings for the 32 unseen ROOD codes by learning any shared embedding components of the VSA when training on other codes. It is not possible for unstructured embeddings to learn better representations for codes never seen in training. We did not see significant differences on the

Table 4: Fine-tuning mean test scores and standard deviations for disease prediction on ROOD task, for both the 6 patients whose codes are completely removed, and the set of patients whose records contain one or more ROOD codes. The best scores are bolded.

| Finetuning Dataset | Configuration | Accuracy | Precision | Recall | F1-Score |
|---|---|---|---|---|---|
| 6 Patients - All Unseen Codes | HRRBase | **94.9**±1.0 | **83.5**±4.6 | **76.8**±5.1 | **79.5**±4.9 |
| | Unstructured | 92.3±0.3 | 46.2±0.0 | 50.0±0.1 | 48.0±0.1 |
| 32k Patients - Partially Unseen Codes | HRRBase | **81.9**±0.1 | 78.3±0.3 | **75.2**±0.8 | **76.4**±0.5 |
| | Unstructured | 81.9±0.2 | **78.7**±0.7 | 74.4±1.2 | 76.0±0.8 |

overall ROOD dataset, where the 32 codes account for only a small percentage of all codes within the 30k records in this testing dataset.

## A.4    More on Ontologies

An ontology formally specifies a conceptualization (i.e., an abstract, simplified view) of a body of knowledge (Gruber, 1995). An ontology defines classes and their relationships and associates these with natural-language labels and descriptions (Gruber, 1993). Hundreds of ontologies spanning all domains of biological and biomedical research have been developed (Kulmanov et al., 2020).

The Systematized Nomenclature of Medicine - Clinical Terms (SNOMED CT) is a comprehensive and mature ontology in the medical domain. It is used in more than 50 countries (Chang & Mostafa, 2021). It defines multi-hierarchical relationships for each concept, and additional type-specific relationships, such as finding site and morphology. Related to our work, similarity measures based on SNOMED CT have been used to cluster documents to support systematic reviews (Chang & Mostafa, 2021).

International Classification of Diseases 9/10 (ICD-9/10) is another disease classification system that is widely used, especially for billing. Although it is well suited to this purpose, it lacks some distinctions that are clinically important, and defines only hierarchical relationships between concepts (Butte & Chen, 2013). We mapped ICD codes to SNOMED CT to take advantage of the latter's richer knowledge representation.

## A.5    Analyzing HRR Configuration Learnable Embeddings

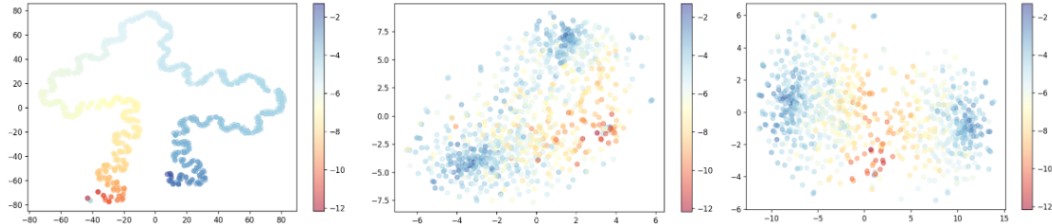

Figure 7: t-SNE representation of sinusoidal frequency embeddings (left), and unstructured embedding components of HRRAdd (middle) and HRRCat (right).

As shown in Figure 1, adding code-frequency information to the structured HRRBase embeddings eliminated their deficiency versus unstructured embeddings. The HRRCat and HRRAdd embeddings, which had unstructured components, outperformed unstructured embeddings. This suggests that their unstructured parts may have learned frequency information, among other things. To investigate whether this occurred, we performed t-SNE dimension reductions of these unstructured components and colored the points by code frequency (Figure 7). The left panel shows a t-SNE reduction of sinusoidal code-frequency embedding for reference, with a clear separation between low and high-frequency codes. The other panels show that the unstructured components of HRRCat and HRRAdd also separate low and high-frequency codes, confirming that code-frequency information was present in these embeddings.

### A.6 ALGORITHMS FOR CONSTRUCTING HRR EMBEDDINGS

In testing algorithms for constructing the HRR embeddings on GPU, we ran into two challenges. First, each concept vector might require a different number of atomic concepts to construct it. Parallelizing the creation of all the concept vectors using default PyTorch operations is not possible, as selecting a different number of atomic vectors per concept vector is not supported with regular PyTorch tensor indexing rules. Naively constructing the concept embedding matrix by building one concept vector at a time is extremely slow as expected, requiring over 9 seconds to do a forward pass and over a minute to perform a backward pass. This amount of time spent creating the embedding matrix would not be practical for training a transformer model end-to-end, we desired a more efficient approach.

We also attempted to use sparse matrix multiplication to essentially perform index lookups and sums of all atomic vectors so that the result could be bound with a relationship vector. This is shown in Algorithm 1. We can view the set of knowledge graphs as a sparse binary tensor, where $\mathbf{G} \in \{0,1\}^{N_c \times N_r \times N_a}$, where $\mathbf{G}_{i,j,k}$ is 1 when a relationship between an atomic symbol and an ICD concept exists. Slicing $\mathbf{G}$ along the relationship dimension yields $N_r$ matrices $\boldsymbol{M}_j : \{0,1\}^{N_c \times N_a}$. These matrices map atomic symbols to concepts if they use relationship $j$. Performing the matrix multiplication $\boldsymbol{M}_j \cdot \boldsymbol{A}$ effectively sums together all atomic vectors that use relationship $j$ and leaves zeros elsewhere. This operation results in a matrix $\boldsymbol{X} \in \mathbb{R}^{N_c \times d}$ which can be circularly convolved with the respective relationship vector, $\boldsymbol{r}_j$ and the result can be added to $\boldsymbol{C}$ to accumulate the final concept embeddings.

While Algorithm 1 was appropriately fast in the forward pass, requiring around 300 ms, it ran out of memory during backpropagation. This is because for each relationship, multiplying $\boldsymbol{M}_j$ and $\boldsymbol{A}$ results in a dense $\boldsymbol{X}$, which is quite large but much of the time also has a lot of zero rows. This approach could work for a different VSA configuration with a fewer number of relationships but is unsustainable for larger numbers of relationships, such as our formulation.

---

**Algorithm 1** Sparse matrix multiplication

**Require:** $\mathbf{G}, \boldsymbol{R}, \boldsymbol{A}$
1: $\boldsymbol{C} \leftarrow \text{zeros}(N_c, d)$
2: **for** $j \in \mathbb{B}$ **do**
3:     $\boldsymbol{r}_j \leftarrow \boldsymbol{R}_{j,:}$
4:     $\boldsymbol{M}_j \leftarrow \mathbf{G}_{:,j,:}$
5:     $\boldsymbol{X} \leftarrow \boldsymbol{M}_j \cdot \boldsymbol{A}$
6:     $\boldsymbol{X} \leftarrow \boldsymbol{r}_j \circledast \boldsymbol{X}$
7:     $\boldsymbol{C} \leftarrow \boldsymbol{C} + \boldsymbol{X}$
8: **end for**

---

Our chosen approach still attempts to combine many operations to contribute to many concept vectors all at once, without having sparse intermediates. We recognize that since $N_r \ll N_c$, many concepts share the same relation. In addition, the number of times a concept uses a particular relation is relatively low. Thus, for a particular relationship, contributes to a lot of concept vectors with a few vectorized circular convolutions.

To formulate this, if we took $\boldsymbol{M}_j$ and created a jagged array of all the atomic symbols related to concept $i$, the number of times the $i$-th row is non-zero is the multiplicity of relationship $j$ for concept $i$. That is, it represents how many times concept $i$ uses relationship $j$ to combine with different atomic vectors. We can gather the indices of 1st atomic vector used for all relevant concepts into a matrix $\boldsymbol{A}_1 = \boldsymbol{A}[\text{atomics}(\boldsymbol{M}_j, 1)]$. We can then perform the operation $\boldsymbol{X} = \boldsymbol{r}_j \circledast \boldsymbol{A}_1$. We then scatter the contents of $\boldsymbol{X}$ into $\boldsymbol{C}$ and add them to the corresponding concept representations already created. Lastly, we repeat this by increasing the multiplicity and gathering new source atomic vectors and scattering the result to various destination concept vectors. Note that $\boldsymbol{X}$ has a different size depending on the multiplicity, since as multiplicity increases, fewer concepts will have a higher multiplicity. This approach is shown in Algorithm 2.

We also took advantage of the linearity of HRR operations to reduce the number of FFT operations performed during circular convolution. Since $\boldsymbol{x} \circledast \boldsymbol{y} = \mathscr{F}^{-1}(\mathscr{F}(\boldsymbol{x}) \odot \mathscr{F}(\boldsymbol{y}))$, we perform the Fourier transforms only once at the beginning and end of the algorithm, and do the Hadamard product in the

---

**Algorithm 2** Gather-scatter

---

**Require:** $\mathbf{G}, R, A$
1:  $C \leftarrow \text{zeros}(N_c, d)$
2:  **for** $j \in \mathbb{B}$ **do**
3:      **for** $m = 1..\max_{i \in \mathbb{A}}(\text{count non-zero}(\mathbf{G}_{i,j,:}))$ **do**
4:          $i_{\text{scatter}} \leftarrow \text{concepts}(\boldsymbol{M}_j, m)$
5:          $k_{\text{gather}} \leftarrow \text{atomics}(\boldsymbol{M}_j, m)$
6:          $\boldsymbol{A}_m \leftarrow \boldsymbol{A}[k_{\text{gather}}]$
7:          $\boldsymbol{X} \leftarrow \boldsymbol{r}_j \circledast \boldsymbol{A}_m$
8:          $\boldsymbol{C}[i_{\text{scatter}}] \leftarrow \boldsymbol{C}[i_{\text{scatter}}] + \boldsymbol{X}$
9:      **end for**
10: **end for**

---

Fourier domain as binding. Algorithm 3 is the implementation we used in all our models and is able to be written with standard PyTorch operations using tensor indexing, element-wise multiplication, and `torch.fft.rfft` and `torch.fft.irfft` for the 1-D real-valued Fourier transforms. All of these operations support autograd and allow for gradients to be computed for the atomic and relationship vector embeddings, with no custom gradient code required.

---

**Algorithm 3** Gather-scatter in the Fourier domain

---

**Require:** $\mathcal{G}, X_r, X_a$
1:  $C \leftarrow \text{zeros}(N_c, d)$
2:  $\boldsymbol{\Xi}_C \leftarrow \mathscr{F}(\boldsymbol{C})$
3:  $\boldsymbol{\Xi}_R \leftarrow \mathscr{F}(\boldsymbol{R})$
4:  $\boldsymbol{\Xi}_A \leftarrow \mathscr{F}(\boldsymbol{A})$
5:  **for** $j \in \mathbb{B}$ **do**
6:      **for** $m = 1..\max_{i \in \mathbb{A}}(\text{count non-zero}(\mathbf{G}_{i,j,:}))$ **do**
7:          $i_{\text{scatter}} \leftarrow \text{concepts}(\boldsymbol{M}_j, m)$
8:          $k_{\text{gather}} \leftarrow \text{atomics}(\boldsymbol{M}_j, m)$
9:          $\boldsymbol{\Xi}_{A_m} \leftarrow \boldsymbol{\Xi}_A[k_{\text{gather}}]$
10:         $\boldsymbol{\Xi} \leftarrow \boldsymbol{\rho}_j \circledast \boldsymbol{\Xi}_{A_m}$
11:         $\boldsymbol{\Xi}_C[i_{\text{scatter}}] \leftarrow \boldsymbol{\Xi}_C[i_{\text{scatter}}] + \boldsymbol{\Xi}$
12:     **end for**
13: **end for**
14: $\boldsymbol{C} \leftarrow \mathscr{F}^{-1}(\boldsymbol{\Xi}_C)$

---

### A.7 LIST OF 32 ROOD CODES

The following is the list of 32 ROOD codes:

1. G248-10: Other dystonia
2. E8498-9: Accidents occurring in other specified places
3. E9688-9: Assault by other specified means
4. Z681-10: Body mass index (BMI) 19.9 or less, adult
5. 30550-9: Opioid abuse, unspecified
6. R262-10: Difficulty in walking, not elsewhere classified
7. E887-9: Fracture, cause unspecified
8. R471-10: Dysarthria and anarthria
9. 9916-9: Hypothermia
10. E9010-9: Accident due to excessive cold due to weather conditions
11. F10129-10: Alcohol abuse with intoxication, unspecified
12. E8499-9: Accidents occurring in unspecified place

13. R636-10: Underweight

14. 920-9: Contusion of face, scalp, and neck except eye(s)

15. R4182-10: Altered mental status, unspecified

16. 95901-9: Head injury, unspecified

17. 78097-9: Altered mental status

18. F29-10: Unspecified psychosis not due to a substance or known physiological condition

19. Z880-10: Allergy status to penicillin

20. Z818-10: Family history of other mental and behavioral disorders

21. 81600-9: Closed fracture of phalanx or phalanges of hand, unspecified

22. 87341-9: Open wound of cheek, without mention of complication

23. H9222-10: Otorrhagia, left ear

24. Z978-10: Presence of other specified devices

25. G20-10: Parkinson's disease

26. G249-10: Dystonia, unspecified

27. 9100-9: Abrasion or friction burn of face, neck, and scalp except eye, without mention of infection

28. 78906-9: Abdominal pain, epigastric

29. E8889-9: Unspecified fall

30. 30500-9: Alcohol abuse, unspecified

31. G520-10: Disorders of olfactory nerve

32. 8020-9: Closed fracture of nasal bones

## A.8 MAPPING FROM ICD TO SNOMED

We prioritized the following order of the mappings and only performed the first valid mapping in the order. Our mapping procedure first mapped ICD-9 to SNOMED CT through the ICD-9-to-SNOMED CT direct map, then ICD-9 to SNOMED CT by the OMOP map. Next, we mapped ICD-10 to SNOMED CT through the OMOP map. The remaining ICD-10 concepts were mapped to ICD-9, and if there existed a valid ICD-9 mapping, that was applied to the ICD-10 concept. This allowed us to maintain the benefit of the more precise ICD-10 codes while also still keeping those that did not have a direct map to SNOMED CT.

The resulting mappings were mostly one-to-one with 19.1% of concepts being one-to-many. For one-to-many mappings, we used all SNOMED CT symbols from all mapped concepts in the VSA representations described below. With OMOP, 96.5% of ICD-9/10 codes have valid SNOMED CT mappings, which is a 19% increase from 77.6% when only using ICD-9/10 to SNOMED CT mappings.

