# OpenReview forum: "Encoding Ontologies with Holographic Reduced Representations for Transformers"
_ICLR.cc/2024/Conference — Submitted to ICLR 2024_

### Official Review · Reviewer_oJkP · 2023-10-22

**Soundness:** 3 good
**Presentation:** 3 good
**Contribution:** 2 fair
**Rating:** 3
**Confidence:** 4

**Summary:**

This article addresses the limitations observed in pre-trained language models (PLMs) when generating high-quality embeddings for seldom-encountered terms, especially within the training dataset. With a particular emphasis on the medical realm, the researchers enrich PLMs by integrating medical ontologies and their associated embeddings.

A central feature of this study is the application of Holographic Reduced Representations (HRR) to encode medical ontologies like SNOMED CT into distinct concept vectors. This is achieved by adding an HRR layer to the transformers found in BERT, allowing for the efficient processing of clinical diagnosis codes.

Empirical results indicate that the enhanced HRR-integrated BERT model delivers superior performance compared to its standard counterpart, particularly in the context of rare diseases.

**Strengths:**

This study delves into the intriguing concept of integrating ontological knowledge with pre-trained language models built on transformer layers. Such an approach holds significant promise and will likely captivate those looking to modify or enrich Large Language Models (LLMs) with additional knowledge in future research.

**Weaknesses:**

- **Methodological Novelty**: While the paper introduces Holographic Reduced Representations (HRRs) into the training of transformer layers, it doesn't clearly articulate innovative techniques or strategies for their integration with transformers and the special challenges into dealing with medical ontologies.

- **Experimentation**: The presented experiments seem insufficient to conclusively establish the superiority of the proposed methods. As evidenced by Table 1 and Fig. 1, the various BERT adaptations demonstrate comparable performances. Furthermore, there's a noticeable absence of comparisons with other foundational benchmarks within the disease diagnosis domain.

- **Reproducibility**: To faithfully replicate the study's findings, more detailed information is necessary. Specifically, clarity on the dataset employed, the design of the architectural models, and the implementation specifics of all the variant baselines would be indispensable.

**Questions:**

- Fig. 1: Why does the HRRBase have such a noticeably lower score compared to the other methods? Does this suggest that HRRBase is the least effective?

- Table 1: While the performances of all methods seem to be on par, what distinguishes HRR from the unstructured approach in terms of advantages?

- Figure 2: Why is there no visualization for the control method, 'Unstructured'? Given that the embedding spaces in the current Figure 2 appear quite analogous, what's the primary takeaway or message from this representation?

---

> ### Author Response · Authors · 2023-11-22
> **Author response to Reviewer oJkP**
>
> Thank you for your review. We appreciate your feedback and suggestions, which we have considered in updating the manuscript.
>
> **Methodological novelty:**
>
> With respect to methodological novelty, we have clarified relevant sections of the paper, particularly the appendix. Creating VSA vectors does not lend itself well to efficient GPU matrix operations, since each concept vector requires a different number of atomic vectors to be combined, i.e. this would be a jagged array over the entire matrix. Earlier attempts at backpropagating through the VSA resulted in running out of memory or required an impractical amount of time to do backprop. We analyzed the operations we wanted to perform and identified where we could use matrix operations with multiple concept vectors to take advantage of the GPU. We also considered that HRR operations can be performed in the Fourier domain, simplifying the computational graph for circular convolution and reducing memory burden during backpropagation. Finally, we used only basic PyTorch methods and did not need to write custom GPU kernels. With these improved efficiencies, this method could scale to a greater number of relationships. We use up to 110 relationships in some more complex variations of our VSA with only 3.5 GB of VRAM.
>
> **Reproducibility:**
>
> Regarding reproducibility, we have improved clarity regarding both the mapping of ICD to SNOMED and data preprocessing. Furthermore, we have provided supplementary materials which contain all code necessary for implementation. The code is properly documented and will be released open source on publication of the manuscript. We aim with our open source code to eliminate any ambiguity that remains in the manuscript.
>
> **Experimentation:**
>
> We have added additional evaluations on the multi-center eICU dataset to A.1. As data from eICU cannot be ordered sequentially, we were not able to apply disease prediction to the dataset. Mortality prediction applied on the eICU dataset has shown performance improvements consistent with our previous experiments.
>
> Furthermore, we have added results of additional experiments in A.2 and A.3 covering rare codes and completely unseen codes. These new results more clearly illustrate the advantages of our model, which are mainly in generalizing to codes that are less common in the training data. In A.2, we show that our HRR embeddings models outperform unstructured models on a disease prediction task for rare codes. In A.3, we show that HRR embedding models outperform unstructured models given a disease prediction task applied on a dataset containing codes that were unseen during training. More details are included in those relevant sections where we show a clear advantage of our HRR methods over unstructured embeddings.
>
> We agree that our method does not perform dramatically better than the standard unstructured embeddings, except in extremes of rare tokens, where we have now shown a strong advantage. However, we believe the work is of interest for several additional reasons. First, while modest, we report a number of statistically significant positive results. Second, independent of performance improvements, our method improves explainability by disentangling representations of code frequency, ontological information, and description. Token frequency in particular bias transformers in ways that may be valuable to check on a decision-by-decision basis, as our method allows.
>
> Finally, although this is likely a controversial perspective, we believe our method is well motivated, so that the performance results are of scientific interest independent of whether they are positive. To reiterate briefly, the motivation is: a) There is a severe imbalance between frequencies of different codes, leading to some embeddings being poorly trained. b) Meanwhile, the code meanings are well defined within ontologies. c) Finally, vector-symbolic architectures provide a well-established method of representing such information. For these reasons, it seems inevitable to study the use of HRRs to provide transformer embeddings with this structured knowledge, so the performance outcome should be of interest even if it were negative (which it is not) and/or the method lacked explainability advantages (which it does not).
>
> *Author response to Reviewer oJkP questions continued below.*

---

> > ### Author Response · Authors · 2023-11-22
> > **Author response to Reviewer oJkP, continued**
> >
> > *Continuing from the Author response above*
> >
> > Answering your questions in order:
> >
> > **Fig. 1: Why does the HRRBase have such a noticeably lower score compared to the other methods? Does this suggest that HRRBase is the least effective?**
> >
> > It suggests that HRRBase is the least effective in terms of the MLM pre-training metric of accuracy, although performance on fine-tuning tasks is of more interest. However, the main take-away from this is the comparison of HRRBase with HRRFreq. HRRFreq is identical to HRRBase except that it explicitly encodes token-frequency information, and it performs on par with unstructured embeddings. As we now explain in Section 3.1, this suggests that the MLM performance difference is due to frequency information, which (as shown in Figure 2) is prominently represented in the unstructured embeddings.
> >
> > Embedded code frequency information can help models make predictions and classifications. However, code frequency information is not medically causal, so using such information to aid medical prediction may not be desirable. At least, the influence of frequency should be explicit. Our HRRFreq method allows toggling the inclusion and exclusion of frequency information to understand its influence on the model’s predictions.
> >
> > **Table 1: While the performances of all methods seem to be on par, what distinguishes HRR from the unstructured approach in terms of advantages?**
> >
> > Table 1 and the associated statistical analysis shows that there are modest performance increases with our proposed methodology, particularly for mortality prediction. Our new experiments show additional situations where HRR outperforms unstructured embeddings, including a strong advantage when the input consists of codes that were unseen in training.
> >
> > Overall, the key advantages are improved generalization to codes that are unseen in training, and explainability, particularly with respect to code-frequency effects.
> >
> > There are additional differences in representation that may provide further practical advantages, although further work is needed to understand these:
> > - HRR embeddings produce broader distributions of cosine similarities
> > - HRR embeddings do not learn a co-occurrence bias that is common in other methods of learning medical embeddings.
> >
> > **Figure 2: Why is there no visualization for the control method, 'Unstructured'? Given that the embedding spaces in the current Figure 2 appear quite analogous, what's the primary takeaway or message from this representation?**
> >
> > We do include visualizations for ‘unstructured’, there are 4 graphs in figure 2, (a) unstructured, (b) HRRAdd, (c) HRRCat, and (d) HRRBase. Graphs (b), (c), and (d) are quite analogous but appear different from (a).
> >
> > In (a) unstructured, we can see that there is a larger gap between frequent and infrequent codes when compared to (b), (c), and (d) which demonstrates that a frequency bias is learned in the unstructured approach that wasn’t learned in any of the HRR approaches. Frequency bias is not desired in the embedding of medical codes as it does not respect medical similarity/reasoning. Medical codes that have similar embeddings should also be medically similar with respect to human knowledge.

---

### Official Review · Reviewer_Q1WN · 2023-11-02

**Soundness:** 3 good
**Presentation:** 2 fair
**Contribution:** 2 fair
**Rating:** 3
**Confidence:** 3

**Summary:**

This paper proposes a neural-symbolic approach (HRRBERT) for transformers by integrating medical ontologies represented by Holographic Reduced Representations (HRR) embeddings. The methodology involves efficiently constructing vector-symbolic embeddings enabling autograd functionality in PyTorch. The experiments demonstrated that the proposed method represents ontological similarities of the codes better than the learned embeddings of the transformer model, and the method can learn similar embedding vectors for medical codes with similar medical meanings.

**Strengths:**

* Results show that the proposed method improves over the baselines, and the t-SNE visualizations demonstrate that codes with similar frequencies are presented close together.
* It is good to make the method compatible with PyTorch autograd.

**Weaknesses:**

* The performance improvements in the fine-tuning task are not very large.
* The authors claim in their paper that the proposed method is efficient. However, there is no direct comparison with the baselines regarding efficiency, thus making the authors' claims that their method is efficient seem a bit unsupported.
* Evaluation is only performed on the MIMIC-IV dataset collected from a single medical center. Further evaluation of multi-center datasets such as eICU could present further information on the method's usefulness.
* The presentation of the paper needs to be significantly improved. The abstract is too long, and while the details described in the main paper are meaningful, they sometimes break the flow of the paper, making the contents difficult to read. I believe the paper needs a significant overhaul to reach a publishable quality for ICLR.
* (minor) Grammar mistakes and typos can be seen in the paper.

**Questions:**

Given the small performance boosts demonstrated in Table 1, I wonder under what circumstances we need to adopt the proposed method.

---

> ### Author Response · Authors · 2023-11-17
> **Presentation of the paper**
>
> We are working to address your comments. Regarding presentation of the paper, we have shortened the abstract, and we have recruited two new arms-length people to review the manuscript for clarity and flow and are incorporating minor changes they have suggested. However, they did not raise significant concerns. If you could provide one or two examples that illustrate this concern we would appreciate it.

---

> > ### Comment · Reviewer_Q1WN · 2023-11-18
> >
> > Thank you for informing me about your progress. For this paper's presentation, one of the issues for this paper is the long abstract. My personal experience informs me that ICLR rarely has such long abstracts in its published works. It is good that you are working on addressing this issue.
> >
> > As for the other issues, my concerns are similar to the other reviewers'. For instance, some citations are missing, such as those for MIMIC-IV dataset. Expressions like 1-to-1 and 1-to-many are also not commonly seen. People often use one-to-one and one-to-many instead. For the main body of the paper, Section 2 contains details that are more appropriate to be discussed in Appendix, such as the method for train-test splits, number of data samples, percentages...etc. I understand that some of them do contribute to the understanding of the ideas in this paper, but it is often good to keep the paper concise since this can allow the reader to interpret the core methodology of the paper rather than setup details. If the readers are interested in further looking into those details, they can check them in Appendix instead. The mapping method discussed in Section 2.2.1 is also not very clear to read to me. The reason behind this approach is not very well-explained.
> >
> > Lastly, I would like to kindly inform the authors that the presentation of the paper does not outweigh the first three weaknesses I mentioned. My weight on the presentation of this paper is much less than those for the results and experiments.

---

> ### Author Response · Authors · 2023-11-22
> **Author response to Reviewer Q1WN**
>
> Thank you for your review. We appreciate your feedback and suggestions. As per your primary suggestions, we have added additional evaluations on the multi-center eICU dataset to A.1. As data from eICU cannot be ordered sequentially, we were not able to apply disease prediction to the dataset. We performed mortality prediction on the eICU dataset with the MIMIC-trained model. There is a distribution shift because eICU counts mortality is defined in a different time window than MIMIC. However, our method showed a small performance improvement. Due to the time constraint, we could not redo the entire training process and analysis with eICU. MIMIC has advantages for our analysis because it has more patients with longer sequences of visits. However, further work with eICU would be a good direction for the future.
>
> Furthermore, we have added results of additional experiments in A.2 and A.3 covering rare codes and completely unseen codes. These new results more clearly illustrate the performance advantages of our model, which are mainly in generalizing to codes that are less common in the training data. In A.2, we show that our HRR embeddings models outperform unstructured models on a disease prediction task for rare codes. In A.3, we show that HRR embedding models outperform unstructured models given a disease prediction task applied on a dataset containing codes that were unseen during training. More details are included in those relevant sections where we show a clear advantage of our HRR methods over unstructured embeddings.
>
> In terms of the presentation of the paper, we have addressed all other reviewers’ concerns. Please see the corresponding replies for details. Additionally, we have shortened the abstract and made edits to methodology sections to improve clarity and flow, taking into account detailed feedback from two colleagues who had not previously read the paper.
>
> Addressing your question:
>
> **Given the small performance boosts demonstrated in Table 1, I wonder under what circumstances we need to adopt the proposed method.**
>
> Our methods should be applied in use cases with imbalanced datasets and where frequency bias is undesirable. Given the existence of frequency bias in the unstructured embeddings, it is unclear how frequency bias learned by the unstructured model contributes to correct or incorrect predictions in fine-tuning tasks. The embeddings produced by our method are less dominated by frequency (as shown in the tSNE plots, Figure 2). As frequency is not a desirable bias in clinical settings, reducing frequency bias in medical AI models may improve trust.
>
> Relatedly, the method should also be used when it is important to understand a network’s reliance on token frequency in a particular decision. Our method disentangles frequency information from semantic information. Transformers have a well-known frequency bias, and we have shown that frequency representation is prominent in learned embeddings. Our method incorporates token-frequency information in a structured way. Although frequency information can be important for accurate predictions (as we have shown by comparing HRRBase to HRRFreq), the frequency information in our embeddings can be added, removed, and modified as needed to examine the role of frequency on individual decisions by the network, enhancing its explainability. In principle, other elements of our embeddings could be treated in a similar way, e.g. to determine the impact of a condition’s category vs. its site in the body. We acknowledge that these important points were unclear in our original submission.

---

### Official Review · Reviewer_fGNU · 2023-11-06

**Soundness:** 3 good
**Presentation:** 3 good
**Contribution:** 3 good
**Rating:** 6
**Confidence:** 4

**Summary:**

This  presents a novel approach for embedding medical concepts into deep learning models using Holographic Reduced Representations. This method, which leverages structured domain knowledge from medical ontologies, enhances the Transformer model's capability to handle rare medical terms and accuracy in predictive tasks. The topic matter holds potential for impactful advancements.

**Strengths:**

Originality:
The authors deserve commendation for their creative integration of medical concepts into deep learning, which is an absolute strength of this paper. By leveraging Holographic Reduced Representations (HRR) to embed structured medical knowledge, they have potentially enhanced the robustness and practical significance of the methodology. This novel use of HRR in the context of deep learning for medical applications sets a new benchmark and opens up avenues for more sophisticated and nuanced models that could transform medical data analysis.

Quality:
The research quality is commendably moderate, detailing the experimental procedures and significance testing methods. However, while it engages in beneficial discussions, there is room for deeper exploration and more rigorous analysis to elevate the robustness of the findings.

Clarity:
The paper is well-structured, with a clear exposition of the problem, methodology, and results. The authors articulate the limitations of current transformer models in processing rare medical terms and effectively convey how their approach addresses these issues. The use of HRR is explained with sufficient detail to be understood by readers who may not be familiar with this representation method.  Particularly noteworthy is the authors' adept use of simple, relatable examples to elucidate the more abstract concepts involved, such as the Holographic Reduced Representations (HRR). This approach significantly aids in demystifying the intricate process for readers who may not be inherently familiar with these techniques.

Significance:
The method introduced in this paper augments the capability of deep learning models to handle intricate medical data, holding promise for the creation of more precise diagnostic tools. This approach has potential implications for the progress of personalized medicine.

**Weaknesses:**

1. The manuscript could benefit from additional explanations of abbreviations such as "SNOMED CT." A glossary or expanded definitions on first use would aid comprehension, especially for readers unfamiliar with the terminology.

2. References to critical resources like the "MIMIC-IV dataset" are absent. Citing such resources would provide context and allow readers to assess the relevance and applicability of the data.

3. Section 2.1 lacks detailed descriptions of dataset handling procedures, which is not reader-friendly for those not acquainted with the "MIMIC-IV dataset." Providing more detail would enhance reproducibility and understanding.

4. Section 2.2.1 does not adequately explain the rationale behind the chosen ICD mapping approach. Specifically, it is unclear why the authors did not map all ICD-10 codes directly to ICD-9 codes and subsequently to SNOMED CT, but rather employed the separate mapping method described. This decision is perplexing and warrants a thorough justification to understand the advantages or the necessity of the approach taken.

5. The choice of employing a "one-sided Dunnett’s test" in Section 3.2 is not substantiated with an explanation. It is essential for the authors to clarify this methodological decision, especially since the results in Table 1 suggest an overlap in the confidence intervals and a close proximity of mean values between the experimental and control groups. The absence of such a justification leaves readers questioning the appropriateness of the statistical test used in the analysis.

6. While Section 3.3 commendably visualizes "highly frequent codes" and "infrequent codes," the lack of separate predictive performance displays for these two categories in Table 1 is a missed opportunity. This differentiation is, after all, one of the paper's stated goals.

7. Finally, the paper does not sufficiently discuss the limitations of the proposed method. Acknowledging and addressing potential shortcomings would strengthen the paper by providing a balanced view and suggesting directions for future research.

**Questions:**

After a thorough review of your manuscript, I have compiled a list of questions and suggestions that I believe could enhance the clarity, completeness, and robustness of your study. Addressing these points may significantly improve the manuscript and aid readers in fully understanding your contributions.

1. The manuscript frequently uses the abbreviation "SNOMED CT" without providing a full explanation or definition for readers who may be unfamiliar with the term. Could you please provide a brief description of "SNOMED CT" and its relevance to your work in the introduction or the first instance where it is mentioned?
2. Dataset Citation:The "MIMIC-IV dataset" is a crucial element in your research, yet it lacks a proper citation or reference. Could you please add a citation for the dataset to allow readers to trace the source and potentially reproduce your study?
3. Dataset Processing Details: In Section 2.1, the description of how the dataset was processed is somewhat brief. Providing a more detailed account of the preprocessing steps would be beneficial, especially for readers who are not familiar with the dataset. Could you elaborate on this process?
4. Rationale Behind Mapping Method: Section 2.2.1 does not sufficiently explain the reasoning behind the chosen ICD mapping method. Why did you opt for the particular mapping approach used in the study instead of mapping all ICD-10 codes to ICD-9 codes, and then to "SNOMED CT"? Clarifying the rationale behind this decision would be helpful.
5. The choice of a "one-sided Dunnett’s test" in Section 3.2 requires further explanation. Could you elaborate on the reasons for selecting this test and whether your method adheres to the assumptions of the test? Additionally, if the decision to use Dunnett's test was deliberate, why opt for a one-sided test instead of a two-sided test? Do you have supporting rationale to support this test? The clarity on these points is crucial, especially in light of the overlapping confidence intervals and closely aligned mean values between the experimental and control groups as presented in Table 1.
6. While the visualization of "highly frequent codes" and "infrequent codes" in Section 3.3 is commendable, the performance metrics under these two categories are not separately displayed in Table 1. Displaying results across various scenarios could effectively showcase the effectiveness of your approach, especially since addressing the representation of infrequent codes is one of your stated objectives. Could you include these performance metrics in your results?
7. A discussion on the limitations of the proposed method is noticeably absent. An acknowledgement of potential limitations and constraints of your approach would provide a more balanced view and contribute to the paper’s integrity. Could you add a section discussing these aspects?

---

> ### Author Response · Authors · 2023-11-22
> **Author Response to fGNU**
>
> Thank you for your review. We have carefully read your comments and have incorporated many of your suggestions to improve the manuscript.
>
> Answering your questions in order:
>
> **1. The manuscript frequently uses the abbreviation "SNOMED CT" without providing a full explanation or definition for readers who may be unfamiliar with the term. Could you please provide a brief description of "SNOMED CT" and its relevance to your work in the introduction or the first instance where it is mentioned?**
>
> We’ve added a brief introduction about SNOMED CT in section 1.1 at the end of paragraph 5. More information on SNOMED CT is included in A.1 as well.
>
> **2. Dataset Citation:The "MIMIC-IV dataset" is a crucial element in your research, yet it lacks a proper citation or reference. Could you please add a citation for the dataset to allow readers to trace the source and potentially reproduce your study?**
>
> We’ve added citations to MIMIC-IV in paragraph 3 of section 1.
>
> **3. Rationale Behind Mapping Method: Section 2.2.1 does not sufficiently explain the reasoning behind the chosen ICD mapping method. Why did you opt for the particular mapping approach used in the study instead of mapping all ICD-10 codes to ICD-9 codes, and then to "SNOMED CT"? Clarifying the rationale behind this decision would be helpful.**
>
> We’ve added more detail on dataset handling to section 2.1. Briefly:
> - We used the ICD-9 and ICD-10 diagnostic codes from the diagnosis table from the MIMIC-IV hosp module.
> - We filtered patients who did not have at least one diagnostic code associated with their records.
> - Sequences of codes were generated per patient by sorting their hospital visits by time.
> - Each unique code was assigned a token.
>
>
> **4. Dataset Processing Details: In Section 2.1, the description of how the dataset was processed is somewhat brief. Providing a more detailed account of the preprocessing steps would be beneficial, especially for readers who are not familiar with the dataset. Could you elaborate on this process?**
>
> We’ve moved some of our mapping methodology from section 2.2.1 to the appendix where we’ve added in additional rationale (A.8). Describing the rationale:
> - ICD-10 was not mapped directly to ICD-9 to maintain preciseness of ICD-10 codes while potentially still keeping codes without mapping from ICD-10 to ICD-9.
> - OMOP mapping greatly improved the coverage (i.e. codes that have a valid mapping) of our ICD-to-SNOMED mapping to 96.5% from 77.6% when only using ICD-9/10 to SNOMED mappings.
>
>
> **5. The choice of a "one-sided Dunnett’s test" in Section 3.2 requires further explanation. Could you elaborate on the reasons for selecting this test and whether your method adheres to the assumptions of the test? Additionally, if the decision to use Dunnett's test was deliberate, why opt for a one-sided test instead of a two-sided test? Do you have supporting rationale to support this test? The clarity on these points is crucial, especially in light of the overlapping confidence intervals and closely aligned mean values between the experimental and control groups as presented in Table 1.**
>
> We’ve added reasoning as to why we chose one-sidedness over two-sidedness in paragraph 1 of section 3.2. We chose a one-sided test because we suspected that HRR methods would improve performance above unstructured controls.
> As the Dunnett’s test is based on computing multiple t-tests, the same assumptions apply for both. Our methods follow with assumptions required by both the Dunnett’s test and t-test (i.e. data is continuous, data is representative, homogeneity of variance, and distribution is approximately normal).
>
>
> As Dunnett’s test is designed for multiple comparisons of treatments to a control, it suited our case where we wished to compare our HRR configuration performance to the unstructured control.
> We have also computed t-tests of our results which agree with our Dunnett’s test analysis.
>
>
> **6. While the visualization of "highly frequent codes" and "infrequent codes" in Section 3.3 is commendable, the performance metrics under these two categories are not separately displayed in Table 1. Displaying results across various scenarios could effectively showcase the effectiveness of your approach, especially since addressing the representation of infrequent codes is one of your stated objectives. Could you include these performance metrics in your results?**
>
> Thank you for this suggestion, we have added this analysis to A.2 (and to some extent, A.3) of our manuscript. We were unclear whether our approach would improve prediction of rare codes, as that may require experience with rare codes in training context. It should more directly improve predictions that have rare codes in their input.

---

> > ### Author Response · Authors · 2023-11-22
> > **Author Response to fGNU (continued)**
> >
> > **7. A discussion on the limitations of the proposed method is noticeably absent. An acknowledgement of potential limitations and constraints of your approach would provide a more balanced view and contribute to the paper’s integrity. Could you add a section discussing these aspects?**
> >
> > We’ve added discussions of limitations into section 4. The first limitation is the complexity of creating HRR compatible knowledge graphs from ontologies. The second limitation is that the HRR method is still 1.5x slower compared to unstructured. Finally, our method relies on rare-code HRRs sharing atomic elements with common codes. However, rare codes are also likely to contain rare atomic elements. This is the likely explanation for separation of rare and common codes in the tSNE reduction of our HRRBase embeddings, which do not contain unstructured elements or explicit frequency information. This limitation suggests a way to improve the method in the future, by incorporating additional information about rare atomic elements (e.g. initializing with pretrained language embeddings of the description words).

---

> > > ### Comment · Reviewer_fGNU · 2023-11-22
> > >
> > > Thanks for your prompt response, which has clarified certain issues (such as Question 4). Nevertheless, my concerns regarding Questions 5 and 6 persist. Without a direct and sufficiently detailed answer to these, I am unable to ascertain the effectiveness of the method. I understand that due to time constraints, it may be challenging for the authors to make further improvements. Therefore,I will keep my evaluation unchanged. Overall, your work has made a valuable exploration.

---

> > > > ### Author Response · Authors · 2023-11-22
> > > >
> > > > Thank you as well for reading this right away. Could you please clarify what remains unaddressed re. #5?

---

> > > > > ### Comment · Reviewer_fGNU · 2023-11-22
> > > > >
> > > > > For example, you say 'we suspected that HRR methods would improve performance above unstructured controls', but this statement is not sufficiently supported unless there is a strong prior. There can be significant differences between the results of one-sided and two-sided tests. Your one-sided test p-value is already close to 0.05, and the differences are minimal as seen in Table 1, which suggests that extra caution is warranted in the hypothesis testing step. Secondly, you claim that you have followed the assumptions required for the method, yet I have not observed the detailed evidence in the revised manuscript that would lead to this conclusion of 'following the assumptions'.

---

> > > > > > ### Author Response · Authors · 2023-11-22
> > > > > >
> > > > > > Thank you for the clarification about the prior assumption. We will report the two-sided Dunnett’s results in the manuscript, which we are working on now. We note the following changes in significance as a result:
> > > > > >
> > > > > > Here are our significant results when we use the 1-sided Dunnett's test:
> > > > > > - HRRBase > Unstructured for MIMIC-IV Mortality Prediction: precision and f1
> > > > > > - HRRBase > Unstructured for MIMIC-IV Disease Prediction: accuracy and precision
> > > > > > - HRRBase > Unstructured for eICU Mortality Prediction: accuracy
> > > > > >
> > > > > > Here are our significant results when we use the 2-sided Dunnett's test:
> > > > > > - HRRBase $\neq$ Unstructured for MIMIC-IV Disease Prediction: accuracy and precision
> > > > > > - HRRBase $\neq$ Unstructured for eICU Mortality Prediction: accuracy
> > > > > >
> > > > > > Additionally, we evaluated the assumptions of the Dunnett's test.
> > > > > > - All groups of variances per metric per experimental configuration were tested with Levene's test. The results suggest that the variances are equal within these groups upon which we applied Dunnett's test.
> > > > > > - We used the Shapiro-Wilk test to evaluate normality of each metric under each configuration. This test suggested 42 / 48 measures meet the normality assumption except for the following:
> > > > > >     - HRRAdd for MIMIC-IV Disease Prediction: accuracy
> > > > > >     - HRRBase for MIMIC-IV Mortality Prediction: precision and f1
> > > > > >     - HRRAdd for eICU Mortality Prediction: accuracy
> > > > > >     - HRRBase for eICU Mortality Prediction: accuracy, recall, precision

---

> > > > > > > ### Author Response · Authors · 2023-11-23
> > > > > > >
> > > > > > > We wanted to report the full summary of statistically significant results presented in this paper. Additionally, the manuscript has been updated to reflect evidence meeting the assumptions for these statistical tests.
> > > > > > >
> > > > > > > - Mortality and Disease prediction fine-tuning tasks using two-tailed Dunnett’s test:
> > > > > > >   - HRRBase has significantly greater accuracy and precision compared to Unstructured for MIMIC-IV Disease Prediction
> > > > > > >   - HRRBase has has significantly greater accuracy compared to Unstructured for eICU Mortality Prediction
> > > > > > > - ROOD disease prediction inference using two-tailed, independent t-test, unequal variance:
> > > > > > >   - HRRBase has significantly greater accuracy, precision, recall, and F1 scores compared to Unstructured for Disease Prediction task on patients with completely unseen codes
> > > > > > > - Top 4 codes by cosine similarity produced by HRRBase embeddings for rare codes are significantly associated with clinical relevance by Fisher’s exact test
> > > > > > > - Frequency-binned MLM top-k accuracy results using two-tailed Dunnett’s test:
> > > > > > >   - HRRBase has significantly lower top-10 and top-100 accuracy compared to Unstructured for common code bins 0, -2, -4, and -6
> > > > > > >   - HRRAdd has significantly greater top-10 accuracy compared to Unstructured for common code bin -2 and rare code bins -8 and -10
> > > > > > >   - HRRCat has significantly greater top-10 accuracy compared to Unstructured for rare code bin -8
> > > > > > >   - HRRCat has significantly greater top-10 accuracy compared to Unstructured for rare code bin -10
> > > > > > >   - HRRAdd has significantly greater top-100 accuracy compared to Unstructured for rare code bins -8, -10, and -12
> > > > > > >   - HRRAdd has significantly greater top-100 accuracy compared to Unstructured for rare code bins -8, -10, and -12
> > > > > > >   - HRRAdd has significantly greater top-100 accuracy compared to Unstructured for rare code bins -10, and -12

---

### Meta-Review · Area_Chair_9ZKd · 2023-12-05

**Metareview:**

This paper presents Holographic Reduced Representations (HRR) for use in deep learning models, specifically for NLP tasks in the medical field. HRR enhances transformer models by embedding medical ontologies, thereby improving performance in tasks such as mortality and disease prediction, and offering better representation for rare medical terms. While the paper is interesting in various aspects, the reviewers have raised reasonable concerns regarding its methodological novelty, experimental design, and comparisons with existing methods. It is hoped that the reviewers' feedback will aid the authors in refining and strengthening the paper.

**Justification For Why Not Higher Score:**

The two responsible reviewers remained unconvinced after considering the rebuttal.

**Justification For Why Not Lower Score:**

N/A

---

### Decision · Program_Chairs · 2024-01-16

Reject